# PRINCIPAL PROTOTYPE ANALYSIS ON MANIFOLD FOR INTERPRETABLE REINFORCEMENT LEARNING

## ABSTRACT

Recent years have witnessed the widespread adoption of reinforcement learning (RL), from solving real-time games to fine-tuning large language models using human preference data significantly improving alignment with user expectations. However, as model complexity grows exponentially, the interpretability of these systems becomes increasingly challenging. While numerous explainability methods have been developed for computer vision and natural language processing to elucidate both local and global reasoning patterns, their application to RL remains limited. Direct extensions of these methods often struggle to maintain the delicate balance between interpretability and performance within RL settings. Prototype-Wrapper Networks (PW-Nets) have recently shown promise in bridging this gap by enhancing explainability in RL domains without sacrificing the efficiency of the original black-box models. However, these methods typically require manually defined reference prototypes, which often necessitate expert domain knowledge. In this work, we propose a method that removes this dependency by automatically selecting optimal prototypes from the available data. Preliminary experiments on standard Gym environments demonstrate that our approach matches the performance of existing PW-Nets, while remaining competitive with the original black-box models.

## 1 INTRODUCTION

Deep reinforcement learning (RL) models have achieved state-of-the-art performance in domains such as Go Silver et al. (2016), Chess Silver et al. (2017), inverse scattering Jiang et al. (2022), and self-driving cars Kiran et al. (2021). More recently, RL has been successfully applied to align large language models with human preferences, receiving considerable attention as a powerful post-training strategy using extensive human feedback data Ouyang et al. (2022); Rafailov et al. (2024). However, despite these advances, the deployment of RL agents in sensitive domains remains limited due to the opaque nature of their decision-making processes. Extracting the rationale behind an agent's actions in a human-interpretable format remains a significant challenge, yet doing so is crucial for understanding failure modes and ensuring trust in these systems. To address this challenge, prototype-based networks have emerged as a promising approach for enhancing the interpretability of deep learning models. ProtoPNet Chen et al. (2019), initially proposed for image classification tasks, introduced pre-hoc interpretability by associating predictions with learned prototype representations.

This idea was later extended to deep RL with Prototype-Wrapper Networks (PW-Nets) Kenny et al. (2023), which provide post-hoc interpretability while preserving the performance of the underlying black-box agent. By incorporating exemplar-based reasoning, PW-Nets allow users to inspect and understand the agent's actions through user-defined reference examples, without degrading task performance. Despite these recent advantages, there is a remaining challenge to automatically and efficiently discover data-adaptive reference examples for interpreting RL behaviors, since manually curated prototypes present several limitations: Human-selected prototypes are costly to acquire, difficult to scale, and often lack consistency across environments, reducing the reproducibility and generalization of explanations. To overcome the above limitations, we propose our principal prototype analysis on manifold: an automated prototype sampling method that eliminates the need for manual intervention and selects prototypes adaptive to RL tasks on the data manifold. To the best of our knowledge, we are the first to automate prototype discovery in RL while retaining the performance

of the black-box agent. Our approach leverages a combination of metric and manifold learning objectives to select prototypes directly from the encoded state space that reflects a low-dimensional geometric representation of the RL task, providing a more scalable and principled mechanism for prototype discovery.

- **Automated and Decoupled Prototype Discovery:** Our method proposes a novel two-stage architecture that decouples prototype discovery from policy optimization. In the first stage, it automatically selects prototypes from the agent's trajectory data using a lightweight neural network trained with combined manifold and metric learning objectives, removing the need for human-curated examples. In the second stage, these prototypes are fixed and integrated into the PW-Net for interpretable action prediction, preserving black-box performance.

- **Geometry-Aware and Faithful Prototypes via Real Instances:** Instead of learning abstract embeddings, our method grounds each learned proxy vector in real training samples by mapping them to their nearest encoded instance. This ensures prototypes are both geometry-aware—by leveraging piecewise-linear manifold approximations—and semantically faithful, enabling more intuitive and interpretable behavior analysis of RL agents.

## 2 RELATED WORKS

Interpretability in neural network architectures, particularly in computer vision (CV) and natural language processing (NLP), has advanced substantially, encompassing both pre-hoc and post-hoc strategies. In CV, post-hoc methods such as Grad-CAM Selvaraju et al. (2019), RISE Petsiuk et al. (2018), and occlusion-based techniques like Meaningful Perturbations Fong & Vedaldi (2017) have enabled visual explanations by highlighting image regions most influential to predictions. However, these methods provide explanations only after decisions are made, offering limited insight into the decision-making process itself. In NLP, pre-hoc approaches include interpretable rule-based decision sets Lakkaraju et al. (2016) and, more recently, Proto-LM Xie et al. (2023), which embeds prototypical reasoning directly into large language models. Post-hoc methods such as LIME Ribeiro et al. (2016) and Integrated Gradients Sundararajan et al. (2017) are widely used to approximate local model behavior and attribute predictions to input features. Other efforts have challenged conventional practices; for instance, Jain & Wallace (2019) questioned the reliability of attention weights as explanations, while Arras et al. (2016) applied Layer-wise Relevance Propagation to trace decision origins in text classifiers.

Although several interpretability techniques have been proposed for reinforcement learning (RL) models Vouros (2022); Milani et al. (2022), most prior work relies on interpretable surrogate models, such as decision trees, that imitate agent behavior in symbolic domains. These approaches, however, do not scale to complex environments with high-dimensional observations such as high-dimensional pixel-based observations. In deep RL settings, most interpretability research has focused on post-hoc methods utilizing attention mechanisms Zambaldi et al. (2019); Mott et al. (2019)or tree-based surrogates Liu et al. (2018), but these often fall short in revealing the underlying reasoning or intent of the agent Rudin et al. (2021). Some approaches attempt to distill recurrent neural network (RNN) policies into finite-state machines Danesh et al. (2021); Koul et al. (2018), but such methods can yield opaque explanations and are constrained to specific architectures.

Our work builds on prototype-based neural networks, which are inherently interpretable by design. These models associate test instances with prototypical examples during the forward pass, enabling intuitive, exemplar-based reasoning. A foundational example of this approach was presented by Li et al. (2017), who introduced a pre-hoc method that learns prototypes in latent space and classifies inputs based on their L2 distance to these prototypes. This method also required a decoder to visualize prototype representations. A notable extension was ProtoPNet Chen et al. (2019), which associated prototypes with image parts rather than entire images, enhancing fine-grained interpretability.

In the RL domain, this concept was adapted by Kenny et al. (2023) through the Prototype-Wrapper Network (PW-Net), a framework that enables pre-hoc performance of the black box model while providing an interpretbale by design post-hoc analysis. The authors also explored an end-to-end learning approach for training prototype representations, inspired by Chen et al. (2019). While effective in image classification tasks, this approach failed to replicate the original performance of

black-box agents when applied to RL environments. We posit that simultaneously optimizing for both performance and interpretability during training introduces a bottleneck that limits effectiveness. To address this, our method decouples these objectives first sampling prototypes using a combination of metric and manifold learning techniques. and then testing the sampled prototypes using PW-net architecture. This separation allows us to preserve the performance of the original agent while maintaining interpretability, without requiring manual prototype selection. Our results show that this strategy achieves competitive performance across multiple environments, highlighting its effectiveness and scalability.

## 3 METHODOLOGY

### 3.1 MOTIVATION

Prototype-based methods offer an interpretable way to associate each class with representative examples; here the representative examples are termed as prototypes. A straightforward baseline to define prototypes is using simple statistics such as the class mean or medoids in the embedding space. However, such naive approaches fail to capture the intrinsic geometry of encoded representations: they are biased by outliers, insensitive to multi-modal distributions within classes, and often yield prototypes that are statistically central but semantically uninformative. To construct meaningful prototypes, it is essential to account for the geometry of the data distribution itself.

According to the manifold hypothesis Cayton (2005), high-dimensional representations typically reside on lower-dimensional manifolds. Leveraging this property enables geometry-aware prototype sampling. Classical manifold learning techniques, however, come with limitations methods like t-SNE van der Maaten & Hinton (2008), UMAP McInnes et al. (2020), and LLE Roweis & Saul (2000) emphasize neighborhood preservation but often distort local dependencies or fail to provide consistent global structure. To address this, we adopt a piecewise-linear manifold learning approach in which nonlinear manifolds are decomposed into locally linear regions. This design ensures that prototypes are drawn from regions that reflect local geometry, avoiding the pitfalls of global averages or distorted embeddings.

While manifold learning preserves geometric structure, prototypes must also be discriminative across classes. Geometry alone does not guarantee that prototypes tightly capture intra-class consistency or maximize inter-class separation. To achieve this, we incorporate a metric learning objectives. Methods such as triplet or contrastive loss require predefined prototypes and extensive sample mining, which is inefficient and often unstable. Instead, we employ Proxy-Anchor loss, which introduces learnable class-level proxy vectors that directly enforce compact clustering within a class and clear separation between classes. After training, each proxy is mapped to its nearest training instance, yielding prototypes that are simultaneously geometry-aware and discriminative.

In Chen et al. (2019), the notion of learnable prototypes was introduced for image classification, where prototype learning was jointly optimized alongside the classification objective. While this approach proved effective for supervised image tasks, its adaptation to reinforcement learning in Kenny et al. (2023) (PW-Net*) resulted in noticeably weaker performance compared to black-box RL models. To overcome this limitation, we propose to decouple these objectives into two sequential stages. In the first stage, we focus on sampling prototypes that serve as robust and representative anchors for each class. In the second stage, these prototypes are fixed and used within PW-Net, which is then trained exclusively on the RL objective.

### 3.2 DATASET

Our method begins with the assumption that we have access to a pre-trained policy $\pi_{\text{bb}}$ operating within a Markov Decision Process (MDP) Sutton & Barto (2015).Since all policies used in our experiments are implemented as neural network architectures, we assume that each policy concludes with a final linear layer. Under this setting, the policy $\pi_{\text{bb}}$ can be decomposed into two components: an encoder $f_{\text{enc}}$, which maps the input state $s$ to a latent representation $z$, and a final linear layer defined by weights $W$ and bias $b$. The resulting policy function can be expressed as:

$$\pi_{\text{bb}}(s) = W f_{\text{enc}}(s) + b,$$

Where $z = f_{\text{enc}}(s)$ represents the encoded state.To construct the dataset used for training our prototype selection mechanism, we execute the pre-trained agent in its original environment for $n$ time steps. During this rollout, we collect encoded state–action pairs, resulting in a dataset $D$:

$$D \leftarrow \{(z_i, \pi_{\text{bb}}(s_i))\}_{i=1}^n.$$

## 3.3 TRAINING OVERVIEW

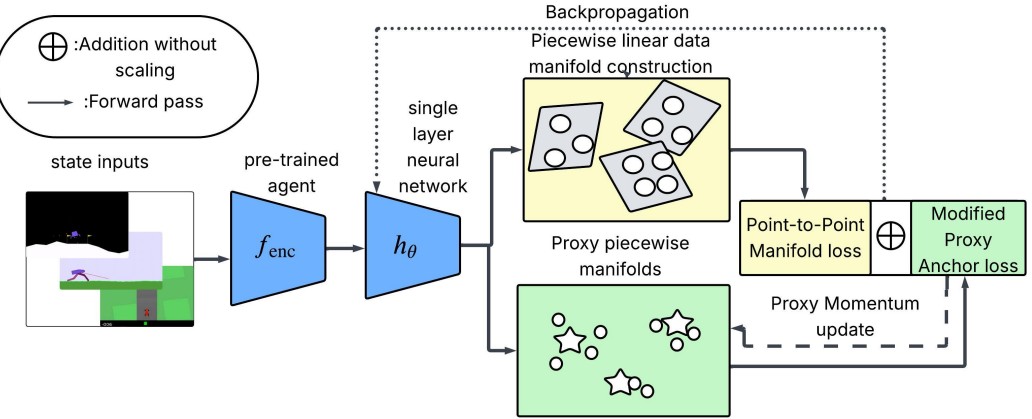

Figure 1: Overview of the proposed method

As mentioned in 3.1, our method consists of two stages. In the first stage of our method, we initialize a simple neural network $h_\theta$ and train it on Dataset $D$ to jointly optimize manifold learning 4 and metric learning objectives 2. The neural network $h_\theta$ learns to map the high-dimensional encoded representations into lower dimensions. Before the training process, we initialize the proxies $\theta_q$ and $\theta_m$; here both the proxies are unique for each class and initiated randomly with $\theta_q = \theta_m$. The proxy vector $\theta_q$ is learned using the metric learning objective 2 and updated via back-propagation. The proxy vector $\theta_m$ is updated via the Momentum update He et al. (2020) where $\gamma$ is the momentum constant.

$$\theta_m \leftarrow \gamma\theta_m + (1 - \gamma)\theta_q \tag{1}$$

Before training our model $h_\theta$, we reformat the dataset $D$ to consist of pairs of encoded state representations and their corresponding discretized actions (Section 4.1). This discretization allows the use of a metric learning objective 2 that clusters encoded states with similar actions and separates those with dissimilar ones, and also enables learning discriminative prototypes.

During training, for every mini-batch $B$ we build linear piecewise manifolds as outlined in 3.4. For every point in $B$, we then compute the manifold-based similarity following the procedure in 3.5. This similarity measure is used to compute the manifold point-to-point loss $\mathcal{L}_{\text{manifold}}$. At the same time, we compute the Proxy Anchor loss $\mathcal{L}_{\text{PA}}$ using randomly initialized class proxies $\theta_q$ and latent representations $z$ in batch $B$. The final loss is computed as $\mathcal{L}_{\text{total}} = \mathcal{L}_{\text{PA}} + \mathcal{L}_{\text{manifold}}$.

The manifold point-to-point loss is designed to reduce the distance between points lying on the same manifold, thus preserving local geometric structure while increasing the distance between points on different manifolds. In contrast, the Proxy Anchor loss encourages samples from the same class to cluster closer together while pushing samples from different classes further apart; this encourages the discriminative learning of prototypes. For every epoch, the network $h_\theta$ is updated through backpropagation, and the proxy vectors are updated according to the procedure described in 1. Once the training is completed, we use the learned proxy vectors $\theta_m$ to select the nearest training data sample as prototypes for each class to be used in the stage of two of training PW-net; here, for every class, there is only $\theta_m$ being initialized, i.e., we will be only getting one prototype per class.

## 3.4 MANIFOLD CONSTRUCTION

Based on the Manifold hypothesis, we assume that the encoded state representations produced by the policy $\pi_{\text{bb}}$, though inherently complex and non-linear, can be locally approximated into smaller chunks of linear regions. Our approach leverages this structural assumption to automatically identify representative prototypes that capture the essential characteristics of each action class.

To efficiently approximate the structure of the data manifold, we adopt a piecewise linear manifold learning method, which constructs localized $m$-dimensional linear submanifolds around selected anchor points. Given a batch $B$ containing $N$ data points, we randomly select $n$ of them to serve as anchors. For each anchor point $h_\theta(z_i)$, we initially collect its $m-1$ nearest neighbors in the encoded representation space based on Euclidean distance to form the neighborhood set $X_i$.

The manifold expansion process proceeds iteratively by attempting to add the $m$-th nearest neighbor to $X_i$. After each addition, we recompute the best-fit $m$-dimensional submanifold using PCA and assess whether all points in $X_i$ can be reconstructed with a quality above a threshold $T\%$. If the reconstruction quality remains acceptable, the new point is retained in $X_i$; otherwise, it is excluded. This evaluation is repeated for subsequent neighbors $N(h_\theta(x_i))_j$ for $j \in \{m_l+1, \ldots, k\}$, gradually constructing a local linear approximation of the manifold.

The final set $X_i$ comprises all points in the anchor's neighborhood that lie well within an $m$-dimensional linear submanifold. A basis for this submanifold is computed by applying PCA to $X_i$ and extracting the top $m$ eigenvectors. We choose PCA for this task as it is computationally efficient and well-suited for capturing linear approximations of non-linear data, in alignment with our assumption of locally linear structure within the high-dimensional state space.

## 3.5 LOSS FUNCTIONS

**Proxy Anchor Loss:** We use a modified version of proxy anchor loss with Euclidean distance instead of cosine similarity:

$$\mathcal{L}_{\text{PA}} = \frac{1}{|\Theta_+|} \sum_{\theta_q \in \Theta_+} \log \left( 1 + \sum_{z \in \mathcal{Z}_{\theta_q}^+} \exp \left( -\alpha \cdot (\|h_\theta(z) - \theta_q\|_2 - \epsilon) \right) \right) \tag{2}$$

$$+ \frac{1}{|\Theta|} \sum_{\theta_q \in \Theta} \log \left( 1 + \sum_{z \in \mathcal{Z}_{\theta_q}^-} \exp \left( \alpha \cdot (\|h_\theta(z) - \theta_q\|_2 - \epsilon) \right) \right) \tag{3}$$

Here, $\Theta$ denotes the set of all proxies, where each proxy $\theta_q \in \Theta$ serves as a representative vector for a class. The subset $\Theta_+ \subseteq \Theta$ includes only those proxies that have at least one positive embedding in the current batch $B$. For a given proxy $\theta_q$, the latent representations $\mathcal{Z}$ in $B$ (where $z \in \mathcal{Z}$) are partitioned into two sets: $\mathcal{Z}_{\theta_q}^+$, the positive embeddings belonging to the same class as $\theta_q$, and $\mathcal{Z}_{\theta_q}^- = \mathcal{Z} \setminus \mathcal{Z}_{\theta_q}^+$, the negative embeddings. The scaling factor $\alpha$ controls the sharpness of optimization by amplifying hard examples when large (focusing gradients on difficult pairs) or smoothing training when small (spreading weight across all pairs). The margin $\epsilon$ enforces a buffer zone between positives and negatives by requiring positives to be closer to their proxies and negatives to be sufficiently farther away.

**Manifold Point-to-Point Loss:** This loss helps in estimating the point to point similarities preserving the geometric structure:

$$\mathcal{L}_{\text{manifold}} = \sum_{i,j} \left( \delta \cdot (1 - s(z_i, z_j)) - \|h_\theta(z_i) - h_\theta(z_j)\|_2 \right)^2 \tag{4}$$

where $s(z_i, z_j)$ is the manifold similarity computed as:

$$s(z_i, z_j) = \frac{s'(z_i, z_j) + s'(z_j, z_i)}{2}$$

with $s'(z_i, z_j) = \alpha(z_i, z_j) \cdot \beta(z_i, z_j)$, where:

$$\alpha(z_i, z_j) = \frac{1}{(1 + o(z_i, z_j)^2)^{N_\alpha}}$$

$$\beta(z_i, z_j) = \frac{1}{(1 + p(z_i, z_j))^{N_\beta}}$$

$\delta$ is the scaling factor, it determines the maximum separation between dissimilar points. The loss encourages Euclidean distances in the embedding space to match manifold-based dissimilarities $1 - s(z_i, z_j)$, ensuring that the learned metric space respects the underlying manifold structure. $o(z_i, z_j)$ is the orthogonal distance from point $z_i$ to the manifold of point $z_j$, and $p(z_i, z_j)$ is the projected distance between point $z_j$ and the projection of $z_i$ on the manifold. The parameters $N_\alpha$ and $N_\beta$ control how rapidly similarity decays with distance, with $N_\alpha > N_\beta$ ensuring that similarity decreases more rapidly for points lying off the manifold than for points on the same manifold..

**Distance Calculation.** For each point pair $(z_i, z_j)$, the distances $o(z_i, z_j)$ and $p(z_i, z_j)$ are calculated using the manifold basis vectors $P_j$ associated with point $z_j$. The projection of $z_i$ onto $P_j$ is computed as $\text{proj}_{P_j}(z_i) = z_j + \sum_k \langle z_i - z_j, v_k \rangle v_k$, where $v_k$ are the basis vectors of $P_j$. The orthogonal distance is then $o(z_i, z_j) = \|z_i - \text{proj}_{P_j}(z_i)\|_2$, and the projected distance is $p(z_i, z_j) = \|\text{proj}_{P_j}(z_i) - z_j\|_2$. This process is repeated for all point pairs, capturing the full geometric structure of the data manifold.

The total loss is the sum of these two components, allowing the model to simultaneously learn a metric space that respects action classes while preserving the geometric structure of the data.

### 3.6 PERFORMANCE REVIEW

The action output $a'$ from the Prototype-Wrapper Network (PW-Net) can generalize better than the original black-box model's action $a$ Snell et al. (2017); Li et al. (2021), due to improved alignment with class-representative prototypes—even without further interaction with the environment. This generalization is critically influenced by the quality and representativeness of the selected prototypes. The black-box policy $\pi_{bb}$ computes the action as:

$$a = W f_{\text{enc}}(s) + b$$

where $z$ is the latent state representation obtained from the encoder. PW-Net enforces structured reasoning through prototypes and computes similarity scores as:

$$a'_i = \sum_{j=1}^{N_i} W'_{i,j} \text{sim}(z_{i,j}, p_{i,j})$$

The similarity function is defined as:

$$\text{sim}(z_{i,j}, p_{i,j}) = \log\left(\frac{(z_{i,j} - p_{i,j})^2 + 1}{(z_{i,j} - p_{i,j})^2 + \epsilon}\right).$$

This ensures actions are chosen based on structured prototype distances rather than raw neural activations. The model uses prototype based regularization providing a better generalization by using the learned policy $\pi_{bb}$ as additional input signal. For simplicity assume a deep RL domain with only two actions possible, the action can be computed as $a'$

$$a'_1 = W'_{1,1} \log\left(\frac{d_{1,1}^2 + 1}{d_{1,1}^2 + \epsilon}\right) + W'_{1,2} \log\left(\frac{d_{1,2}^2 + 1}{d_{1,2}^2 + \epsilon}\right)$$

$$d_{i,j} = z_{i,j} - p_{i,j}.$$

Where $W'$ is the manually defined weight matrix for each action, the output $a'$ is heavily dependent on the similarity score between the $z_{i,j}$ and $p_{i,j}$, this metric helps PW-Net avoid completely mimicking the policy $\pi_{bb}$ and instead use it as an additional input signal along with the choice of prototype to better align responses with human choices.

## 4 EXPERIMENTS

### 4.1 ACTION DISCRETIZATION

In continuous action domains, we standardize the action space by first converting all action values to their absolute values. We then apply the sigmoid function to these transformed values and determine the final action label by selecting the index corresponding to the maximum sigmoid output. For instance, in the Car Racing environment, the original action output is represented as a tuple `[(acc, brake), left, right]`. We first restructure this into a unified vector format: `[acc, brake, left, right]`. The encoded state representation is then assigned a discrete label based on the index of the maximum value obtained after applying the sigmoid function to this transformed vector. This discretization procedure is consistently applied across all continuous action environments, including the Bipedal Walker and Humanoid Standup environments, enabling compatibility with our prototype selection and metric learning pipeline.

### 4.2 NUMERICAL RESULTS

| Method | Car Racing (Reward) | Bipedal-Walker (Reward) | Humanoid Stand up (Reward) |
|---|---|---|---|
| Our method | **220.91 $\pm$ 0.85** | 312.10 $\pm$ 0.17 | **75112.60 $\pm$ 840.25** |
| PW-Net | 220.72 $\pm$ 0.34 | 308.27 $\pm$ 3.41 | 74980.37 $\pm$ 816.84 |
| VIPER | N/A | -89.71 $\pm$ 7.51 | - |
| PW-Net* | -9.48 $\pm$ 2.50 | 190.41 $\pm$ 59.51 | - |
| k-means | -2.09 $\pm$ 0.94 | -107.72 $\pm$ 0.13 | - |
| Black-Box (DQN) | 219.56 $\pm$ 0.85 | **312.32 $\pm$ 0.21** | 74930.50 $\pm$ 837.61 |

Table 1: Reward comparison on Car Racing, Bipedal Walker, and Humanoid Standup tasks

| Method | Pong (Reward) | Lunar Lander (Reward) | Acrobat (Reward) |
|---|---|---|---|
| Our method | **14.96 $\pm$ 0.45** | **218.01 $\pm$ 1.47** | **-83.12 $\pm$ 2.39** |
| PW-Net | 10.72 $\pm$ 0.26 | 216.38 $\pm$ 1.69 | -84.67 $\pm$ 2.42 |
| VIPER | N/A | -408.81 $\pm$ 60.98 | - |
| PW-Net* | 8.85 $\pm$ 1.69 | 124.54 $\pm$ 120.53 | - |
| k-means | -21.00 $\pm$ 0.00 | -419.46 $\pm$ 119.08 | - |
| Black-Box | 12.07 $\pm$ 0.39 | 214.75 $\pm$ 1.08 | -85.54 $\pm$ 3.37 |

Table 2: Reward comparison on Pong, Lunar Lander, and Acrobat environments.

The PW-Net Kenny et al. (2023) relied on human-curated prototypes in visually interpretable environments such as Car Racing. However, this approach becomes infeasible in complex domains with high-dimensional, non-visual state spaces and large continuous action sets. For instance, the Humanoid Standup environment B.1 features a high-dimensional vector input and 17 continuous control actions across joints and rotors, making manual prototype selection impractical without domain-specific tools or expertise. Our automated prototype selection method overcomes this limitation by leveraging geometric and class-level structure in the latent space. Notably, in the Humanoid Standup task, our approach achieves a mean reward of 75,112.60 (SE = 840.25), closely matching the original black-box model's performance of 74,930.50 (SE = 837.61). This result demonstrates that our method retains performance even in settings where manual prototype curation is infeasible. For the new environments of Humanoid Standup and Acrobat for calculating the results on PW-net, we used the class mean as the prototype. This is in reference with the approach followed by the authors of the PW-net C.1 that they have used for training on the Bipedal-walker and Lunar Lander environments. To analyze the effect of varying hyperparameters, we have performed an ablation study D on the Bi-pedal and Atari pong environments.

### 4.3 USER STUDY

The interpretability of PW-Nets arises from their case-based reasoning approach, where decisions are explained through analogies to representative prototypical states. Prior work Kenny et al. (2023)

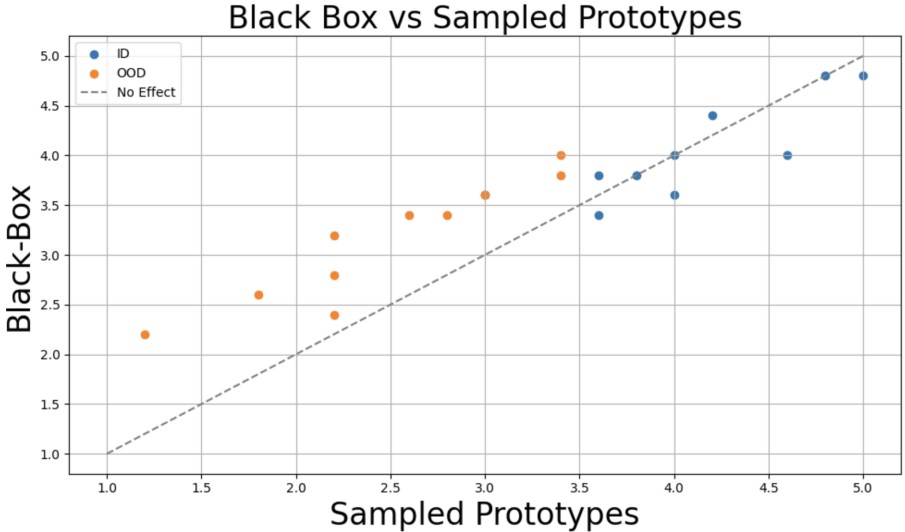

Figure 2: IID and OOD distribution plots for both user groups

demonstrated that human-selected prototypes enable users to form accurate mental models of agent behavior, supporting effective prediction of both successes and failures. Our automated prototype selection is designed to preserve this interpretability mechanism by identifying states that capture the same decision-critical features that human experts would highlight.

To evaluate the plausibility and faithfulness of the sampled prototypes, and to analyze how prototype-based explanations influence participants' ability to interpret and anticipate the agent's decisions in both IID and OOD conditions, we conducted a user study in the CarRacing environment (Figure 6). Out of the six environments considered in our experiments, four are symbolic domains where states are represented as vectors of physical properties, while CarRacing and Atari Pong operate on raw pixels that can be visually interpreted. CarRacing was chosen because its driving actions are naturally understandable to non-expert users, making it suitable for visual inspection and evaluation Rudin et al. (2021).

Two groups of 25 participants were recruited. The first group interacted with PW-Nets using our sampled prototypes as global explanations, while the second group was assigned to a black-box condition in which participants were told: "The car has learned to complete the track as fast as possible in this environment by learning from millions of simulations, but no explanation is available." In this condition, prototype images were replaced with text-only information, while the prototype group received visual exemplars that directly conveyed the agent's reasoning process. This design isolates the contribution of prototypes to interpretability by contrasting a case-based explanation with no explanation.

Participants were presented with 20 scenarios: 10 in-distribution (ID) from the standard CarRacing-v0 environment where the agent drove safely, and 10 out-of-distribution (OOD) from a modified environment NotAnyMike (2025) introducing new road types and red obstacles that led to actual failure cases. After viewing the car's current state and the corresponding explanatory condition, participants predicted whether the vehicle would operate safely on a five-point Likert scale. This setup assessed how well explanations enabled users to anticipate agent behavior in both familiar and novel situations.

Results are summarized in Figure 6. In the ID scenarios, both groups produced similar ratings, indicating that participants could reliably interpret safe behavior in either condition. In contrast, for the OOD cases where the agent failed, participants in the prototype condition were more sensitive to these failures: their ratings more closely reflected the unsafe ground truth, while the black-box group tended to overestimate safety. This demonstrates that prototype-based explanations enhance interpretability by helping users anticipate failure modes, even if they do not increase overall reported confidence.

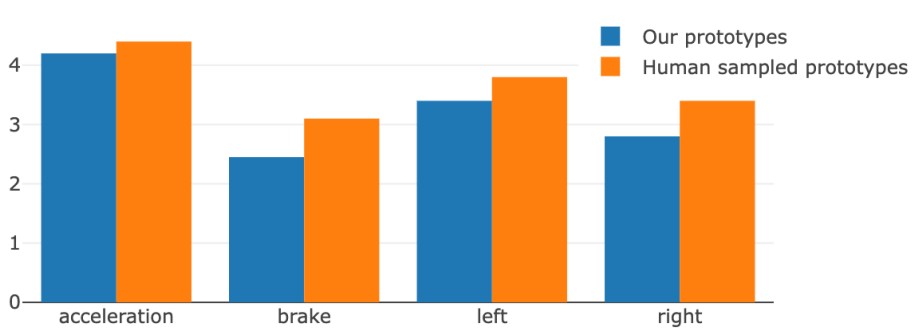

Figure 3: Comparison of Visual similarity between prototypes

In addition, we evaluated the interpretability of our sampled prototypes relative to the human-curated prototypes used in PW-Nets (Figure 4.3). For each action class, participants rated on a 1–5 scale how well the prototype represented the corresponding decision. For the acceleration class, ratings were comparable across both methods, while for the other classes human-curated prototypes were slightly preferred. However, the differences were marginal, suggesting that the automatically sampled prototypes are equally interpretable in practice. Importantly, our method delivers this interpretability benefit in high-dimensional settings where human prototype selection is infeasible.

## 5 CONCLUSION AND FUTURE WORK

The application of Deep Reinforcement Learning (Deep RL) spans from automated game simulations to fine-tuning large language models (LLMs) using preference data. However, in the absence of transparency regarding the agent's actions and intentions, deploying such systems in high-stakes or sensitive domains remains impractical Rudin (2019). PW-Net addresses this challenge by providing interpretability for deep RL agents through example-based reasoning using human-understandable concepts. While relying on human-annotated prototypes offers valuable insights, it is not feasible across all the domains. To overcome this limitation, our approach automatically samples prototypes from the training data itself. Through user studies, we also demonstrate that trust in the model's behavior—especially under out-of-distribution (OOD) scenarios where failures are likely—can be effectively assessed.Nonetheless, our method faces challenges when applied to tasks like sentence generation in LLMs. Specifically, our technique assumes a single prototype per class, which becomes infeasible when the output space is as large as the vocabulary size—potentially in the order of millions. Extracting prototypes at that scale is computationally intensive, requiring methodological adaptations for interpretability in such settings. Although Xie et al. (2023) proposed an extension of prototype learning for LLMs, their work was limited to sentence classification, which does not address the prototype scaling issue in generative tasks.

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

# A RELATED WORKS

## A.1 MANIFOLD LEARNING

The manifold hypothesis is a well-established principle in Machine Learning, which suggests that Cayton (2005):

> *Although data points often appear to have very high dimensionality, with thousands of observed features, they can typically be represented by a much smaller set of underlying parameters. In essence, the data resides on a low-dimensional manifold embedded within a high-dimensional space.*

Based on the Manifold hypothesis Manifold learning focuses on uncovering low-dimensional structures in high dimensional data. Manifold learning techniques like TSNE van der Maaten & Hinton (2008),UMAP McInnes et al. (2020), LLE Roweis & Saul (2000) and Isomap Tenenbaum et al. (2000) utilize information derived from the linearized neighborhoods of points to construct low dimensional projections of non-linear manifolds in high dimensional data.

The method Piecewise-Linear Manifolds for Deep Metric Learning Bhatnagar & Ahuja (2024) aims to train a neural network to learn a semantic feature space where similar items are close together and dissimilar items are far apart, in an unsupervised manner. This method is based on using linearized neighborhoods of points to construct a piecewise linear manifold, which helps estimate a continuous-valued similarity between data points.

## A.2 METRIC LEARNING

Metric learning aims to learn an embedding space where semantically similar samples are close and dissimilar ones are far apart. Common loss functions include **Contrastive loss** Hadsell et al. (2006),aims at making representations of positive pairs closer to each other, while pushing negative pairs further away than a positive margin. It is commonly used in tasks such as face verification or representation learning with Siamese networks. Here $(z_i, z_i')$ are embeddings of a pair, $y_i \in \{0, 1\}$ indicates similarity, and $m$ is the margin.

$$\mathcal{L} = \frac{1}{N} \sum_{i=1}^{N} \left[ y_i \, \|z_i - z_i'\|_2^2 + (1 - y_i) \max \left( 0, m - \|z_i - z_i'\|_2 \right)^2 \right]$$

**Triplet loss** Schroff et al. (2015) is another metric learning objective that enforces relative similarity by ensuring that an anchor $x_a$ is closer to a positive sample $x_p$ (same class) than to a negative sample $x_n$ (different class) by at least a margin. Unlike contrastive loss, which only considers pairwise distances, triplet loss leverages relative comparisons, making it more effective in learning discriminative embeddings for tasks such as face recognition and image retrieval, here $f(\cdot)$ is the embedding function, $m$ is the margin, $x_a$ is the anchor, $x_p$ is a positive sample, and $x_n$ is a negative sample.

$$\mathcal{L} = \frac{1}{N} \sum_{i=1}^{N} \max \left( 0, \, \|f(x_a^i) - f(x_p^i)\|_2^2 - \|f(x_a^i) - f(x_n^i)\|_2^2 + m \right)$$

**Multi-class N-pair loss** Sohn (2016) generalizes triplet loss by comparing one positive sample against multiple negative samples simultaneously. This encourages more efficient optimization than triplet loss, which only considers a single negative at a time, leading to better embedding separation for tasks such as image classification, retrieval, and verification. Here $f(\cdot)$ is the embedding function, $x_a^i$ is the anchor, $x_p^i$ is the positive sample of the same class, and $\{x_n^j\}$ are negatives from other classes.

$$\mathcal{L} = \frac{1}{N} \sum_{i=1}^{N} \log \left( 1 + \sum_{j \neq i} \exp \left( f(x_a^i)^\top f(x_n^j) - f(x_a^i)^\top f(x_p^i) \right) \right)$$

**Supervised contrastive loss** Khosla et al. (2021) extends contrastive loss by leveraging label information to pull together embeddings from all samples of the same class, rather than relying only on pairwise similarity. Unlike contrastive loss, which is limited to positive and negative pairs, supervised contrastive loss uses class supervision to exploit multiple positives per anchor, leading to richer and more discriminative representations. Here $P(i)$ is the set of indices of positives sharing the same class as anchor $x_i$, $\tau$ is a temperature scaling parameter, and $f(\cdot)$ is the embedding function.

$$\mathcal{L} = \sum_{i=1}^{N} \frac{-1}{|P(i)|} \sum_{p \in P(i)} \log \frac{\exp\left(f(x_i)^\top f(x_p)/\tau\right)}{\sum_{a=1}^{N} \mathbf{1}_{[a \neq i]} \exp\left(f(x_i)^\top f(x_a)/\tau\right)}$$

**Proxy-Anchor Loss:** Proxy-Anchor Loss Kim et al. (2020) replaces anchors with learnable class representatives (proxies), removing the need for anchor sampling as in contrastive, triplet, or N-pair losses. Instead of comparing individual samples, embeddings are optimized against proxies, which serve as stable anchors for each class.

$$\mathcal{L}_{\text{PA}} = \frac{1}{|\Theta_+|} \sum_{\theta_q \in \Theta_+} \log \left( 1 + \sum_{z \in \mathcal{Z}_{\theta_q}^+} \exp\left(-\alpha \cdot (s(z, \theta_q) - \epsilon)\right) \right)$$

$$+ \frac{1}{|\Theta|} \sum_{\theta_q \in \Theta} \log \left( 1 + \sum_{z \in \mathcal{Z}_{\theta_q}^-} \exp\left(\alpha \cdot (s(z, \theta_q) - \epsilon)\right) \right)$$

# B    MODEL ARCHITECTURE

This section includes details about the black-box models, user study, and the model architecture ($h_\theta$) used in our method. We used a single-layer network with intermediate normalizations. The prototype size is set to 50 for all the environments. The motivation for using a simpler model is to avoid losing information in the encoded vectors during manifold construction.

Table 3: Model Architecture

| Layer | Layer Parameters |
|---|---|
| Linear | (latent size $z$, prototype size) |
| InstanceNorm1d | prototype size |
| ReLU | - |

## B.1    BLACK-BOX MODELS

For the CarRacing environment, we used a CNN model trained using PPO JinayJain (2025). This pre-trained model was evaluated under both IID and OOD settings during the user study. For Atari Pong, we used a simple CNN trained with the Double Dueling DQN method bhctsntrk (2025). The model used for BipedalWalker was trained using TD3 nikhilbarhate99 (2025b), and the LunarLander model was trained using the Actor-Critic method nikhilbarhate99 (2025a). These networks are relatively simple, reflecting the symbolic nature of their respective environments. For the Humanoid-Standup and CartPole environments, we used models from Stable-Baselines3 Raffin et al. (2021), trained using PPO with an MLP policy. The diversity of environments, models, and algorithms demonstrates the robustness of our approach.

## C    TRAINING PARAMETERS

For the first phase of training—prototype discovery—we train our network for 200 epochs on the training dataset (Section 3.2) using two separate Adam optimizers: one for the network parameters

and one for the proxy parameters. Both optimizers use a learning rate of `1e-3`, accompanied by a learning rate scheduler with decay rate $\eta_t = 0.97$. The dimensionality of the encoded vector $z$ varies depending on the environment and the encoder model, but generally falls near the order of 100. We use a mini-batch size of 128 samples and set the reconstruction threshold $T$ to 90%. The scale parameter $\delta$ is set to 2 (the maximum distance between two points on a unit sphere), and the submanifold dimension $m$ is fixed at 3.

For the second phase—training and evaluating the sampled prototypes within the PW-Net framework—we use the Adam optimizer with a learning rate of `1e-2`, again paired with a scheduler using $\eta_t = 0.97$. Training and evaluation are conducted over 5 independent iterations. In each iteration, the PW-Net model is trained for 10 epochs and evaluated over 30 simulation runs to compute the mean and standard deviation of the resulting rewards.

All experiments were conducted on an NVIDIA RTX A6000 GPU. In the first stage of our method, we train a lightweight neural network $h_\theta$ to sample prototypes, which requires approximately 640 MB of GPU memory and about 7 hours of training time without parallelization. With parallelized estimation of manifold-based similarities, the training time is reduced to roughly 2 hours, with a peak GPU memory usage of about 4700 MB across all environments. For the Humanoid Standup and Acrobat environments, we did not evaluate the methods VIPER, PW-Net*, and k-means, as our focus was on approaches that achieve performance closer to the original black-box model. As observed in the remaining four environments, these methods consistently fall short of delivering results comparable to the black-box baseline.

## C.1 METHOD COMPARISONS

The k-means method selects prototypes by choosing the cluster centers within each action class. In contrast, PWnet* learns prototypes through a joint objective that combines a clustering loss and a separation loss while simultaneously optimizing for RL performance. Moreover, our approach significantly reduces the reliance on subjective inputs, thereby promoting a more objective assessment of the prototypes. For all the environments, we used the same black box models B.1 used in PW-net; consequently, the values of performance for the methods PW-net*, VIPER Bastani et al. (2019), and K-means were also taken from the paper PW-net.

For the Car Racing and Atari Pong environments, we recomputed the performance of the black-box models but retained the PW-Net scores reported in Kenny et al. (2023), as their evaluation relied on human-surveyed prototypes for these tasks. In the case of symbolic domains, we constructed "ideal" prototypical action-space examples, where the action of interest was set to 1 or -1 and all others to 0, and subsequently mapped these to the closest training samples. These prototypes were then used to reevaluate PW-Net's performance across the four symbolic domains in this work.

# D ABLATION STUDY

To analyze the effect of each individual parameter, we have performed an ablation study on one model each from the Continuous and discrete action spaced environments. To achieve this we used the Bi-pedal walker and Atari pong environments respectively.

## D.1 EFFECT OF $m$

The parameter $m$ denotes the dimension of the linear submanifold $X_i$, which locally approximates the data manifold around a point $h_\theta(z)$. To examine its effect, we vary $m$ in the range $[2, 8]$ with a step size of 1. As shown in (Figure 5 and Figure 4)(a), performance consistently decreases in both the environments as $m$ increases. This trend arises because $X_i$ is intended to approximate the immediate neighborhood of a point, which is inherently low-dimensional. Larger values of $m$ may lead to overfitting, since only a limited number of nearby samples are available within a batch to reliably estimate $X_i$, thereby degrading performance. Furthermore, we observe that the computational overhead for prototype sampling increases with larger $m$, underscoring the trade-off between accuracy and efficiency.

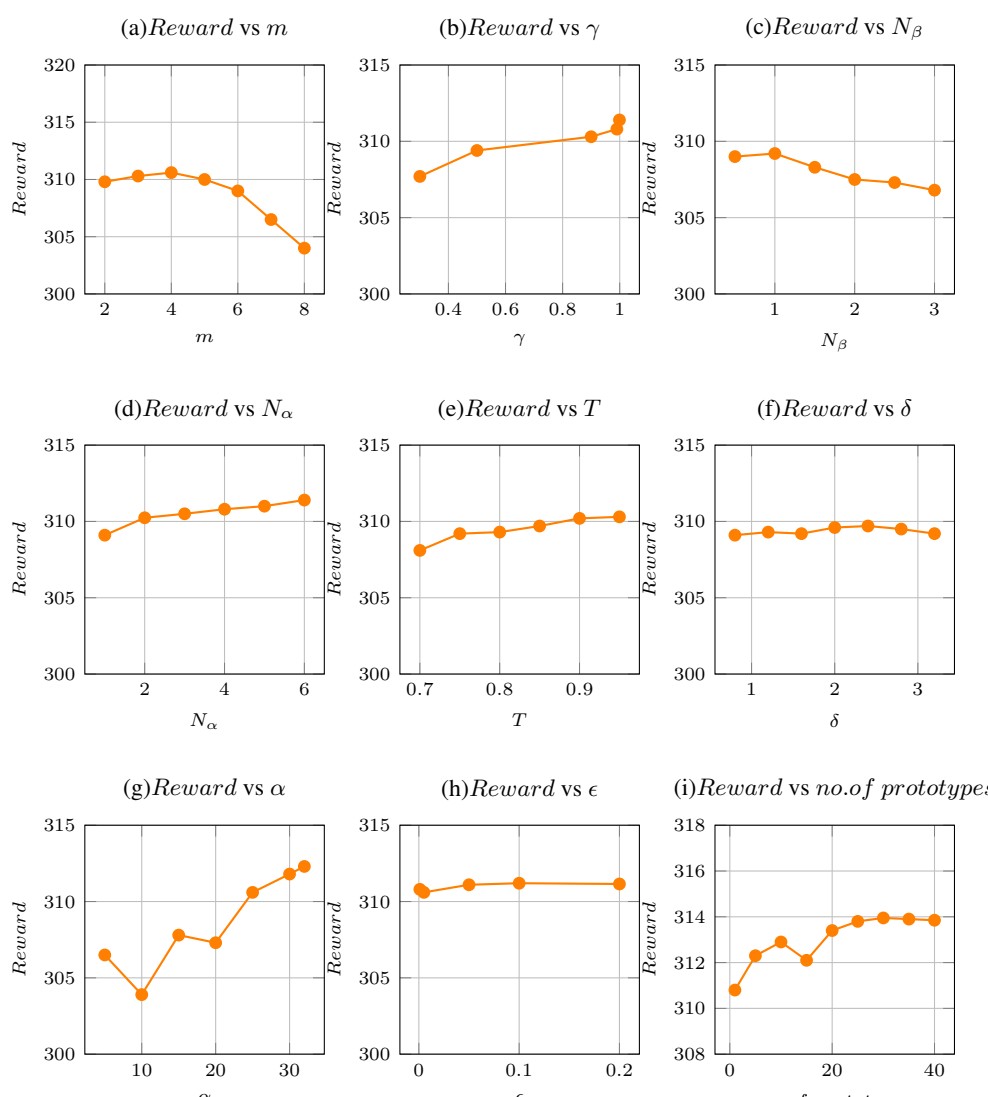

Figure 4: Ablation study on Bi-pedal walker environment

## D.2 EFFECT OF $\gamma$

The parameter $\gamma$ denotes the momentum constant used to update the proxy vector $\theta_m$ during prototype sampling. Following He et al. (2020), higher values of $\gamma$ are expected to yield improved performance, as the proxy updates become smoother and more stable. Consistent with this observation, (Figure 5 and Figure 4)(b) shows that in both models, performance improves as $\gamma$ increases, highlighting the importance of stable momentum updates for effective representation learning.

## D.3 EFFECT OF $N_\alpha$ & $N_\beta$

The parameters $N_\alpha$ and $N_\beta$ control the decay of similarity based on the orthogonal and projected distances, respectively, of a point from the linear submanifold in the neighborhood of another point. We vary $N_\alpha$ in the range $[1, 6]$ with a step size of 1, and $N_\beta$ in the range $[0.5, 3]$ with a step size of 0.5. As shown in (Figure 5 and Figure 4)(c), increasing $N_\beta$ leads to decrease in performance in both the environments. In contrast, (Figure 5 and Figure 4)(d) shows that performance improves with larger $N_\alpha$ in both the environments.

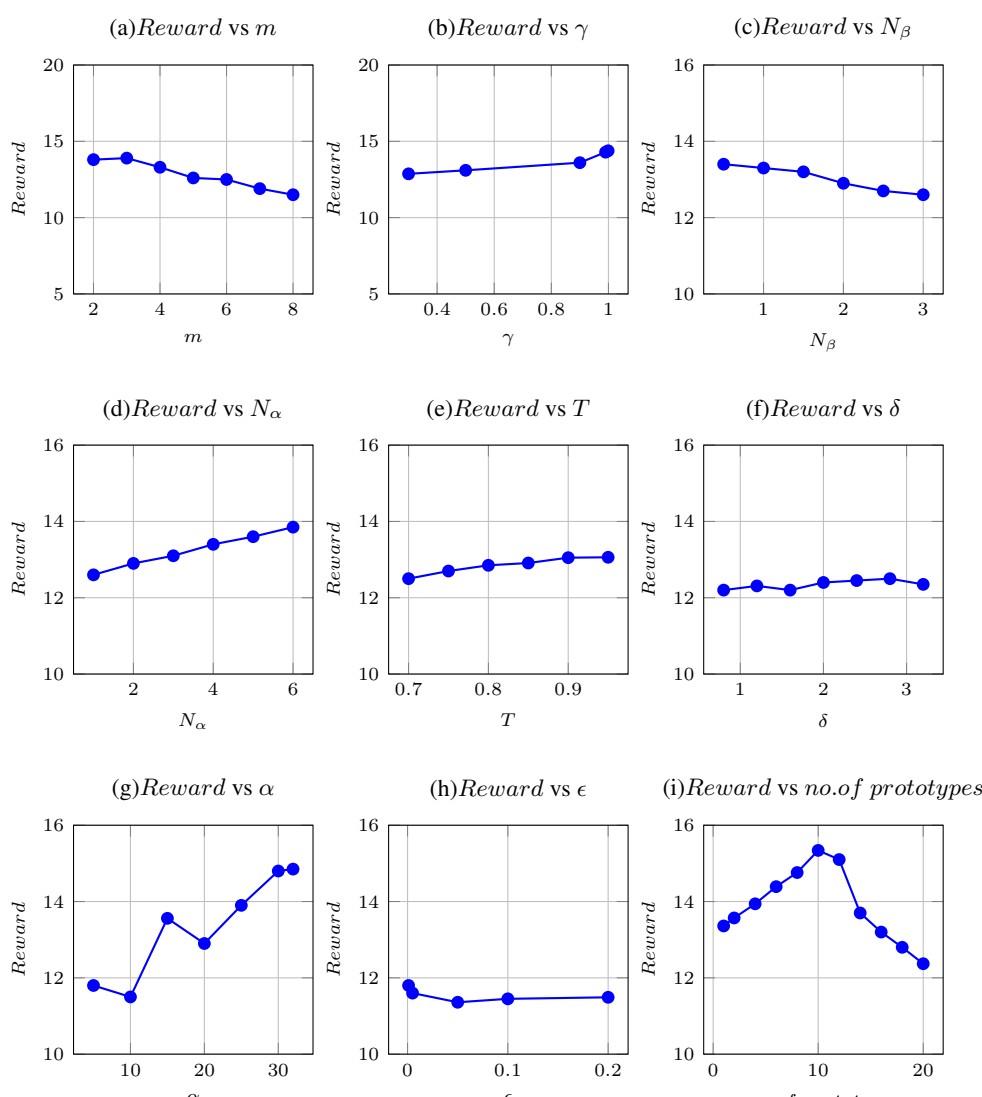

Figure 5: Ablation study on Atari Pong walker environment

This effect can be explained by the relationship between $N_\alpha$ and $N_\beta$: as $N_\alpha$ approaches $N_\beta$, a point $A$ at distance $\varepsilon$ within the linear neighborhood of a point $B$ (and thus sharing many features with $B$ and its neighbors) may be treated as equally dissimilar to $B$ as another point $C$ located at an orthogonal distance $\varepsilon$ from the neighborhood of $B$. In the experiments when $N_\beta$ was varied $N_\alpha$ is set to 4, as $N_\beta$ increases from 0.5 to 3 it becomes closer to $N_\alpha$ which is leading to a decrease in performance. When $N_\alpha$ was varied from 1 to 6 $N_\beta$ was set to 0.5, as $N_\alpha$ increases from it becomes larger than $N_\beta$ which is leading to an increase in performance.

## D.4 EFFECT OF $T$

The reconstruction threshold $T$ determines the quality of points admitted into the linear submanifold $X_i$. We vary $T$ in the range $[0.7, 0.95]$ with a step size of $0.05$. As shown in (Figure 5 and Figure 4)(e), the models in both environments exhibit a clear upward trend in performance as $T$ increases, underscoring the importance of ensuring that only high-quality points are incorporated into $X_i$.

### D.5 EFFECT OF $\delta$

The scaling factor $\delta$ regulates the maximum separation between dissimilar points. We vary $\delta$ in the range $[0.8, 3.2]$ with a step size of $0.4$. As shown in (Figure 5 and Figure 4)(f), the performance remains relatively stable across this range in both environments, highlighting the robustness of our method.

### D.6 EFFECT OF $\alpha$

The scaling factor $\alpha$ controls the sharpness of the exponential term in the Proxy Anchor loss. We vary its value over $5, 10, 15, 20, 25, 30, 32$. As shown in (Figure 5 and Figure 4)(g), models in both environments exhibit an overall increasing trend in performance with larger $\alpha$.

### D.7 EFFECT OF $\epsilon$

The margin parameter $\epsilon$ enforces that positive embeddings are pulled within this distance from their corresponding class proxies. We vary its value across $0.001, 0.005, 0.05, 0.1, 0.2$. As shown in (Figure 5 and Figure 4)(h), models in both the environments demonstrate stable performance across the range of $\epsilon$, undermining its effect in the loss function.

### D.8 EFFECT OF $no.of\ prototypes$

To investigate the effect of prototype count on performance, we conducted an ablation study in the Bipedal Walker and Atari Pong environments. In Bipedal Walker 4(i), rewards consistently increased with additional prototypes until reaching a plateau. In contrast, in Atari Pong 5(i), rewards initially improved with more prototypes but began to decline beyond a certain point. We attribute this divergence to differences in state representation.

Bipedal Walker is a symbolic domain where states encode physical properties such as position and velocity, providing relatively low-noise inputs. By comparison, Atari Pong represents states as raw pixels, which must be encoded by a neural network before prototype selection. This pixel-based encoding introduces noise, and as the number of prototypes increases, the accumulated noise degrades performance.

## E USER STUDY

Two groups of 25 users each participated in the study 6. One group was presented with the black-box model (the "Black-Box Group"), while the other with sampled prototypes (the "Sampled Prototypes Group"). Both groups were given identical scenarios and instructions on how to rate them independently. The figure below shows a sample of the IID and OOD cases shown to users.

## F LLM USAGE

LLM was used to improve the quality of writing, and to assist in the LaTeX code review; it was not used during the ideation or experimentation phase.

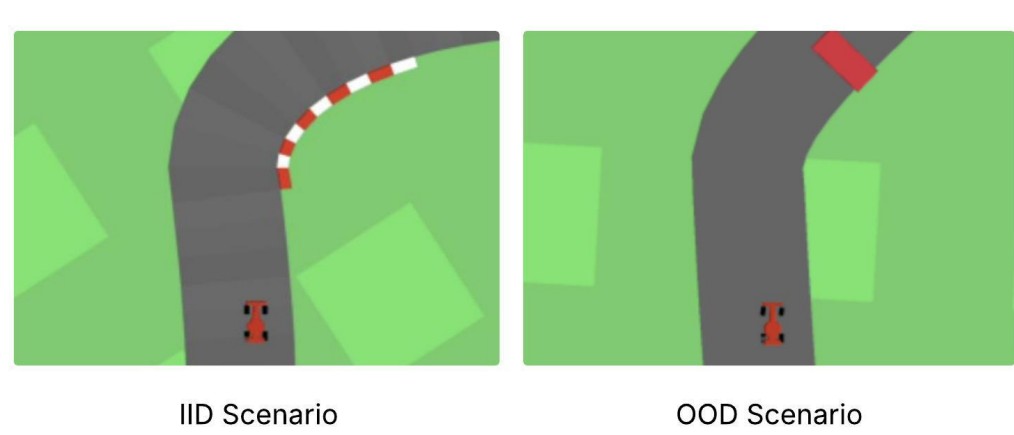

IID Scenario                    OOD Scenario

Figure 6: User Study Overview

