# OpenReview forum: "Principal Prototype Analysis on Manifold for Interpretable Reinforcement Learning"
_ICLR.cc/2026/Conference — ICLR 2026 Conference Withdrawn Submission_

### Official Review · Reviewer_kyfg · 2025-10-29

**Soundness:** 2
**Presentation:** 1
**Contribution:** 2
**Rating:** 2
**Confidence:** 3

**Summary:**

This work aims to generate prototype-based explanations for reinforcement learning tasks using automated prototype discovery. It builds upon prototype-wrapper networks introduced for RL explanation earlier and attempts to improve upon this by automatically finding appropriate prototypes from the training batch of data. According to my understanding, the paper's approach has two parts:  first learning an approximate linear manifold of data using a loss that allows for clear action-based separation of the environment states and then applying the prototype wrapping for generating explanations without a loss in the agent performance. The paper demonstrates experiments on various RL environments to showcase the improvement in performance over existing techniques, primarily the prototype-wrapper net. A human study is also presented wherein the method generated prototypes are used instead of human-annotated prototypes in the earlier work.

**Strengths:**

1) The idea of automatically discovering the prototypes is novel and has high significance for RL explainability.
2) The motivation behind this line of work summarized in line 128:  "According to manifold hypothesis high-dimensional representations typically reside on lower-dimensional manifolds." is solid.
3) The method outperforms other methods on the agent performance preserving aspect.

**Weaknesses:**

**Major: (my reasons for not giving higher score)**
1) No instances of the prototype explanations are provided in the paper.
2) The writing flow is inverted -- section 3.3 talks about training overview by introducing previously undefined terminology and constantly refers to the later sections for disposition. This is unnecessarily increasing the burden in comprehending the core of this paper. The paper heavily relies on the past work on Prototype-Wrapper Networks by Kenny et al [1] for various concepts and unfortunately does not include any definitions or background information about following concepts:   a. Proxy anchor, b. Classes in the reinforcement learning setting. It only gets clear on line 252 that a proxy is supposed to be a representative vector for a class (which I am assuming is a discrete action).

3) Section 3.4 on Manifold Construction: At least, I am unable to follow how the method works due to lack of any geometric/visual aid for understanding the construction.
4) Human study results are supposed to be summarized in Figure 6 as per line 426. However, that figure only shows two frames of car racing environment side by side without any explanatory description provided.
5) Figure 3 has not been referred anywhere in the text as far as my reading goes. There is no sufficient explanation provided either. It does not particularly talk about how the visual similarity is measured and if these similarities are so perceptibly similar for a human that the proposed automated prototype discovery will reliably replace human sampling of prototypes.


**Minor:**
1) The policy \pi_bb(s) is defined as having an affine transformation of the state encoding f_{enc}(s). It would be more correct to also discuss when such a transformation is valid because the \pi_bb(s) has to be a probability distribution defined over the actions available in state "s".
2) Figure 5 caption says "Atari Pong walker environment" which seems to be mixing up "Atari  Pong" and "Mujoco Walker" environments. Also, without multiple randomized trials, it is hard to comment about the statistical significance of this result. Same with figure 4.

**Unclear parts:**
1) Line 192: "Before the training process, we initialize the proxies \theta_q and \theta_m; here both the proxies are unique for each class and initiated randomly with \theta_q = \theta_m". What do you mean by "proxies"?
2) Line 195: "Momentum update". Do you mean "Polyak averaging"? The term momentum update is defined in the context of gradient-based optimization and does not seem to be applicable in the context of the present paper.
3) Line 202: "discriminative prototypes". This discrimination is w.r.t. the actions at a given state or w.r.t. the states for a given action?
4) Line 475: "Although Xie et al. (2023) proposed an extension of prototype learning for LLMs, their work was limited to sentence classification, which does not address the prototype scaling issue in generative tasks." <- this sentence should be part of related work and not the concluding statement of the paper.


**A few critical grammatical errors**
1) Line 110: "objectives first sampling prototypes using a combination of metric and manifold learning techniques. and then" -> "objectives -- first sampling prototypes using a combination of metric and manifold learning techniques, and then"

**Discussion on related work:**
Related works do not include very relevant line of work that attributes RL decisions to past trajectories, e.g.: "Deshmukh, S. V., Dasgupta, A., Krishnamurthy, B., Jiang, N., Agarwal, C., Theocharous, G., & Subramanian, J. (2023). Explaining RL decisions with trajectories. ICLR 2023."


*References:*

[1] Eoin M. Kenny, Mycal Tucker, and Julie Shah. Towards interpretable deep reinforcement learning with human-friendly prototypes. In The Eleventh International Conference on Learning Representations, 2023

**Questions:**

I have asked my questions while discussing perceived weaknesses in the above section.

---

### Official Review · Reviewer_R8ND · 2025-10-29

**Soundness:** 2
**Presentation:** 3
**Contribution:** 2
**Rating:** 2
**Confidence:** 3

**Summary:**

Two-stage pipeline that learns data-grounded prototypes with Proxy-Anchor and local linear manifolds, then plugs a single prototype per action class into a PW-Net wrapper; shows black-box-level returns and a small CarRacing user study.

**Strengths:**

- Removes manual prototype curation by mapping proxies to nearest real states.
- Competitive performance compared to black box methods on several established benchmarks.
- Includes a user study validating interpretability claims.

**Weaknesses:**

- Novelty is limited: recombining PW-Net-style prototype policies [1] with post-hoc/learned-prototype lines [2–4] and standard Proxy-Anchor metric learning [5].
- The manuscript lacks direct head-to-head with ProtoX [2], PW-Net variants [1], and classic RL-explanation baselines such as VIPER [6].
- Its unclear whether this approach is scalable. The method defaults to one prototype per class despite multi-modality; prior work uses multiple prototypes per class/node—report multi-prototype results and scaling [3, 4].
- User study lacks statistical rigor: 5-point Likert only; no power analysis, hypothesis tests, p-values, or confidence intervals [7].

**Questions:**

1) Add requested baselines.
2) Your discretization maps continuous actions to a single class via sigmoid+argmax over [acc, brake, left, right]. How sensitive are results to: number of bins per dimension, binning strategy, and multi-label cases when multiple controls co-activate?
3) What is the effect of increasing prototypes per class on performance and interpretability? Please report where gains plateau [3,4].
4) The user study does not appear to be statistically rhigherous, please follow [7].
5) "Results are summarized in Figure 6." This figure does not appear to contain any information on the study?
6) Has the user study underwent IRB approval?


References\
[1] Kenny, Eoin M., Mycal Tucker, and Julie Shah. "Towards interpretable deep reinforcement learning with human-friendly prototypes." The Eleventh International Conference on Learning Representations. 2023.\
[2] Ragodos, Ronilo, et al. "Protox: Explaining a reinforcement learning agent via prototyping." Advances in Neural Information Processing Systems 35 (2022): 27239-27252.\
[3] Chen, Chaofan, et al. "This looks like that: deep learning for interpretable image recognition." Advances in neural information processing systems 32 (2019).\
[4] Nauta, Meike, Ron Van Bree, and Christin Seifert. "Neural prototype trees for interpretable fine-grained image recognition." Proceedings of the IEEE/CVF conference on computer vision and pattern recognition. 2021.\
[5] Kim, Sungyeon, et al. "Proxy anchor loss for deep metric learning." Proceedings of the IEEE/CVF conference on computer vision and pattern recognition. 2020.\
[6] Bastani, Osbert, Yewen Pu, and Armando Solar-Lezama. "Verifiable reinforcement learning via policy extraction." Advances in neural information processing systems 31 (2018).\
[7] Doshi-Velez, Finale, and Been Kim. "Towards a rigorous science of interpretable machine learning." arXiv preprint arXiv:1702.08608 (2017).

**Details Of Ethics Concerns:**

The manuscript does not mention IRB/ethics approval or review, so it does not appear to have undergone IRB review.

---

### Official Review · Reviewer_3eyb · 2025-11-01

**Soundness:** 2
**Presentation:** 2
**Contribution:** 1
**Rating:** 2
**Confidence:** 3

**Summary:**

# Problem:

Reinforcement Learning (RL) is permeating in all aspects of AI. It is used to train black-box models of increasing complexity. However, it lacks widely-usable explainability methods that do not involve a trade-off between interpretability and performance.

Prototype Wrapper-Networks (PW-Nets) is one example of an explainability method that do not sacrifice performance for interpretability, but it is not widely-usable because it requires expert domain knowledge to define reference prototypes. Reliance on human-selected prototypes makes the approach costly, difficult to scale, and it yields inconsistent explanations across environments.

How to make PW-Nets more widely-applicable, reducing both cost of applicability, scaling difficulties, and inconsistency of the quality of explanations across environment ?

# Contributions:

The paper proposes a method to automatically select prototypes from available data in an optimal fashion, using a principal prototype analysis on manifold.

Evaluation using a user study demonstrate that ‘trust in the model’s behaviour […] can be effectively assessed’ using the proposed automatic prototype sampling method, without loss of performance.

**Strengths:**

## Quality:

SQ1: I appreciate the motivation paragraph for its valuable problem framing insights.

However, I think it could be further improved my providing formal framing of the problem of selecting prototypes from available data, by also highlighting how those prototypes are later use to provide interpretability to human users. I would advise the authors to consider expanding on that.

SQ2: Experiments presenting results on a fairly wide set of environments is appreciated. However, it seems that the user study of Section 4.3 is only performed over the Car Racing environment. This limits the external validity of the claim of the paper with respect to the interpretability of the method. I would advise the authors to perform a user study in at least one, at best all, other environments to strengthen their claim (or show clear insights about the limitations of the method in specific environments). I do appreciate the explanation about using the CarRacing game over the other games though (that is why I am not flagging this as a weakness of the paper).

## Originality:

SO1: To my knowledge, this is the first paper to propose a performance-preserving automatic prototype sampling algorithms for PW-Nets.

## Significance:

S/WS1: I appreciate the inclusion of Figure 3 towards providing a measure of the interpretability of the method’s prototypes. It is indeed necessary for the paper to show internal validity, but I must flag that the evidence is rather poor for the following reasons:

1. there is no statistics about the context of the data (.e.g how many datapoints for each bar? Are bars showing mean or median?) It is important to show the standard deviation or error in order to enable some appreciation of the robustness of the results.

2. better yet, the figure should be accompanied with statistical significance tests (e.g. p-values of test checking whether each of the distributions A vs distributions B are indeed different or similar, depending on your claim)

3. Limitations related to the humans that selected the sampled prototypes: e.g. what is their level of expertise in the domain where the prototypes were sampled? For instance, if they are not expert, then the figure would provide evidence that your proposed method is able to sample prototypes that are better or worse or similar to non-expert human’s prototypes, which does not really aligns with the main claim of the paper (this is an issue of internal validity: your comparison would no longer allow you to investigate what you have set out to investigate - which is whether your method is able to sample prototypes of high quality)

**Weaknesses:**

## Quality:

WQ1: Beyond the introduction, I have found the paper to be overall difficult to follow. While the motivation section and the method are sufficiently well-explained, there remains major issues: crucially, at the beginning of the Experiment section, I would expect details about the baselines and comparison agents used in order to provide context to the results. (e.g. What is VIPER? Why is it included here?) Moreover, none of the tables are referenced in the text, so it is unclear how we should understand the results presented.
Furthermore, in ln426, Figure 6 is referenced as summarizing the results of the user study, but it seems to be a mistake because  it points to some Car Racing environment screenshots in the appendix.

I am assuming it is meant to refer to Figure 2 (please correct me if not?), and in this case I am afraid I do not understand what does it present: what do the x and y axes correspond to? Could they be ratings on a 5-point Likert scale? It might help the readers if you could explain one of the point of the graph, for example. I do not understand what do the points on this graph represent. As a result, I do not understand the explanation text that follows ln426. It might be helpful to clearly states the hypotheses tested and what would be the form of the results that would validate or invalidate those hypothesis, for instance, maybe?
(I can now see Section C.1 in the appendix: I think this should be placed in the main text, or at least explicitly referenced from the main text’s experimental section introduction paragraph, maybe?)

WQ2: The paper does not provide details about the limitations of the proposed approach, beyond a short discussion in the conclusion. I would advise the authors to revise the paper by including measures that tangibly show the limitation of the proposed approach.

WQ3: While I appreciate the inclusion of an ablation study that shows quantitative measures of the reward and how hyperparameters impact them, I would have expected an ablation study with respect to the interpretability aspects of the proposed method, as this is the main claim of the paper (unless I am mistaken, because the claims are not explicitly listed, but I am referring to lns453-458 and lns466-469).

## Clarity:

WC1: The claims of the paper are not explicitly stated, which makes it hard to read the paper and also to appreciate the value of the work. I am assuming that the main claim is that the proposed method is able to sample prototypes that yield interpretability on par with that of expert-humans-sampled prototypes (following lns21-24, 47-53, 456-459).

WC2: Citations are formatted in a way that hurts readability (maybe missing \citep command?).

WC3: Section 3.4 :  I would like to advise the authors to resort to an algorithm to convey those details more effectively. As it stands, those lines are very detailed but lacks an overview at a more coarse level to make the whole section more readable.

WC4: Section 4 lacks an introductory paragraph that would provide the necessary context, and prime the readers for the details to come in the following subsections. e.g. looking at the first subsection, it is not clear to me why are action discretization being discussed in this context?

## Significance:

S/WS2: With its performance-preserving feature, the proposed automatic prototype-sampling algorithm for PW-Nets seems of significance to the greater AI community using RL, and especially when application is in sensitive domains.

However, in the current state of the paper, which lacks many details about the proposed algorithm and experiments (cf. S/WS1 to start with), it is unclear how significant the paper can be.

**Questions:**

Please see Strengths & Weaknesses.

---

### Note · Authors · 2025-12-03

I have read and agree with the venue's withdrawal policy on behalf of myself and my co-authors.